# A Fault Forecasting Approach Using Two-Dimensional Optimization (TDO)

## Abstract

Data preparation plays a pivotal role in every machine learning-based approach, and this holds true for the task of detecting claims in the automotive industry as well. Handling high-dimensional feature spaces, especially when dealing with imbalanced data, poses a significant challenge in sectors where a vast amount of data accumulates over time. Machine learning models trained on highly imbalanced data often result in unreliable and untrustworthy predictions. Therefore, addressing the aforementioned issues is essential during the data pre-processing phase. In this paper, we propose an innovative two-dimensional optimization approach to effectively address the challenge of highly imbalanced data in the context of fault detection. We employ a heuristic optimization algorithm called Genetic Algorithm to concurrently reduce both the data point tuples and the feature space. Furthermore, we constructed and evaluated two-dimensional reduction using particle swarm optimization (PSO) and Whale optimization algorithms. The empirical results of the proposed techniques on the data collected from thousands of vehicles show promise.

**keywords:** Fault Detection, Tuple Selection, Feature Selection.

## 1 Introduction

Modern vehicles in the automotive industry are complex systems with a multitude of potential configurations, where component breakdowns can originate from various sub-components failing due to different reasons. A rise in component breakdowns can indicate a quality issue with the component, which in turn elevates the risk to customer safety, even in modern vehicles, and negatively impacting customer satisfaction. Thus, fault detection has become a critical operation in the maintenance strategy of the automotive sector. In this context, numerous studies utilize various statistical and machine learning algorithms to predict claims in various scenarios Khoshkangini et al. (2020a;b). The current availability of data collected from hundreds of sensors in vehicles provides us with the opportunity to employ this data for usage modeling and claims estimation. Advanced machine learning and artificial intelligence (AI) have become essential in many applications Rabbani et al. (2020; 2016); Dahl et al. (2020); Revanur et al. (2020); Khoshkangini et al. (2017), with particular significance in the automotive sector, where we develop multiple predictive models to predict component breakdowns before they occur.

Nonetheless, this vast volume of data comes with deficiencies that can impact the construction of predictive models. Redundant readouts and features often infiltrate the data, imposing an additional processing burden and resulting in highly imbalanced data. This imbalance can lead to bias in the modeling process and negatively affect prediction performance. Consequently, researchers have given substantial attention to addressing imbalanced data by applying different machine learning and data mining techniques. For instance,Chawla et al. (2002) introduced the Synthetic Minority Over-sampling Technique (SMOTE), which is designed to increase the number of minority instances by interpolating along a line, while ignoring the majority instances. In Chawla et al. (2003), the SMOTE technique is enhanced by integrating it with a boosting procedure. However, despite the notable improvements achieved with SMOTEBoost in addressing the problem, the approach does exhibit vulnerability to artefacts. In the context of the SMOTE technique, Han et al. introduced two methods known as borderline-SMOTE1 and borderline-SMOTE2 in their paper Han et al. (2005).

These methods focus on selectively oversampling only the minority instances that are in proximity to the borderline. In a similar study discussed in Bunkhumpornpat et al. (2009), the authors introduced the Safe-Level-SMOTE technique. This approach involves sampling minority instances along the same line with different weights and considering them for analysis. To enhance its performance, the method synthesizes minority samples more prominently around a larger safe model.

In the study by the authors in P. Songwattanasiri (2010), they introduced the Synthetics Minority Over- and Under-sampling Techniques (SMOUTE). This method combines over-sampling of minority data using SMOTE with under-sampling achieved through k-means clustering. SMOUTE offers advantages such as faster computation and improved F-measure values, particularly beneficial for handling big data, in contrast to the plain SMOTE approach. Bunkhumpornpat et al. Bunkhumpornpat et al. (2011) proposed the Majority Under-sampling Technique (MUTE). This technique involves establishing a boundary between the minority and majority samples, and during the training process, it discards all of the majority data that falls within the minority class boundary.

Regarding the studies discussed in this context, including those mentioned earlier, we observe that the fundamental concept behind under-sampling approaches is to remove the majority samples, which are often regarded as artifacts, with the primary goal of preserving minority cases. This principle can lead to a slight reduction in the performance of predictive models. Moreover, there are numerous studies in the literature that delve into the challenges posed by imbalanced data. The methods discussed earlier have the potential to be applied and further enhanced in the context of the automotive industry, particularly in the domain of breakdown detection. Machine learning approaches, as exemplified in studies such as Khoshkangini et al. (2019); Chaudhuri (2018); Killeen et al. (2019); Hecker et al. (2018); Khoshkangini et al. (2020b), have found utility within the automotive sector for predicting failures and enhancing reliability. Over the past decade, researchers have introduced new approaches to address these challenges, as documented in studies such as Ran et al. (2019); Khoshkangini et al. (2019; 2020a; 2021). Within these investigations, various artificial neural network (ANN) architectures have been explored for tasks such as estimating the remaining useful life of components (RUL), including applications to bearings and wind turbines Teng et al. (2017). Multi-layer perceptrons (MLP) have been employed to predict breakdowns as a classification task Revanur et al. (2020), and linear regressions have been utilized for forecasting downtimes Welch et al. (1995). These studies have garnered considerable attention among researchers in the field.

In response to the challenges outlined above, this study introduces a novel two-dimensional optimization approach aimed at mitigating the issues associated with highly imbalanced data, specifically in the domain of fault detection within the automotive industry. Our proposed system introduces a novel approach that combines optimization techniques with tuple and feature selection methods. In this context, *tuple* refers to the recorded readout samples gathered during the truck's operational lifespan, while *features* encompass the characteristics that describe the behavior of vehicles throughout their operational life. In this study, our proposed methodology focuses on selecting the most informative tuples and features that have a significant impact on the predictive models, enabling accurate predictions of component breakdowns. We employ three types of optimization algorithms: Genetic Algorithm (GA) Whitley (1994), Particle Swarm Optimization (PSO) Marini & Walczak (2015), and Whale Optimization Algorithm (WOA) Mirjalili & Lewis (2016). These algorithms are utilized to select the optimal tuples and the most informative predictors for use in the training phase. It's important to note that in this work, we place a particular emphasis on the application of GA. The objective is to identify which portion of the training data significantly contributes to the predictive model, thereby improving its performance. The optimization process is aimed at extracting a specific subset of the data (tuples and features), that results in the most accurate predictions. Subsequently, the outputs obtained from the three optimization approaches are compared under various conditions to assess their respective efficiencies.

The rest of the paper is organized as follows: In Section 2, we explain the base algorithm used in this study. Data representation and preparation are described in Section 3. In Section 4, the proposed approach is discussed. Section 5 covers experimental results, and the summary is expressed in Section 6.

## 2 BACKGROUND

In this Section, we describe some notions of *Genetic*, *PSO*, and *WOA* algorithms.

### 2.1 GENETIC ALGORITHM (GA)

Genetic is an evolutionary algorithm Dennett & Dennett (1996) suitable for constrained and unconstrained optimization tasks, which is widely used in a vast range of applications Srinivas & Patnaik (1994); Whitley & Sutton (2012); Motieghader et al. (2017); Ma et al. (2018). Unlike the other optimization approaches used in Dorigo & Blum (2005); Khoshkangini et al. (2014), GAs work with a coded representation of the problem data set and look for a population of possible solutions to the problem. GA constantly generates a population of chromosomes as solutions. Through several generations, by using GA operators, the system randomly selects individuals from the current population to be parents and uses them to generate a new population for the next generation. The GA operators are briefly described as follows:

- Encoding: the approach utilized a binary scheme operation. The binary is the most common encoding scheme, where each chromosome $c_i$ is a vector of operators represented as a binary of 1 or 0. In this encoding strategy, each individual feature $f_i$ shows that whether it is included $f_i = 1$ or not $f_i = 0$ in that particular chromosome $C_{(i=1,...,m)}$ Katoch et al. (2020).

- Generation/initialization: The initialization of the population is constructed after the encoding operation. By randomly selecting the individuals, the first population is created with labels of either 1 or 0, While the first indicates the individual predictor and the latter signifies that the predictor is not selected Katoch et al. (2020).

- Gene selection: in this step, different subsets of genes (from the training set) are selected over various iterations. The final subset of the gene will be chosen from the genes with the highest selected numbers Deng et al. (2004).

- Mutation: To maintain the diversity of the genes from one generation to another, the mutation occurs where some of the genes are subjected to mutate with low probability Katoch et al. (2020).

- Crossover: After calculating the suitability of the chromosomes, two children will be produced by exchanging a specific part of the genes of the two chromosomes.

### 2.2 PARTICLE SWARM OPTIMIZATION ALGORITHM

The main idea of the PSO algorithm originated from the collective movement of animals, including birds. Birds usually choose their landing place according to the least danger and the greatest opportunity. The philosophy behind this decision is based on each bird's experience and personal perceptions (pBest) as well as observation of other birds' movements or social knowledge (gBest). In the PSO algorithm, birds are called particles, which are formed randomly. In each phase, the particles occupy a more suitable position in the problem space compared to the previous phase. Their fit is determined by the objective function, similar to the genetic algorithm we discussed in the previous section. Their fit is determined by the objective function, similar to the genetic algorithm. In this study, we used the binary version of the algorithm to solve our discrete problem, while the PSO algorithm is mainly used for continuous problems. In feature selection, Ones and Zeros indicate the presence or absence of the tuples or features. The purpose of optimization techniques is to determine the variable that is represented by the vector $P = [p_1, p_2, p_3, \ldots, p_n]$ and is minimized depending on the formula of the objective function where $n$ represents the number of variables that may be specified in the problem. The position vector in the PSO is calculated by the following formula.

$$P_i^t = [p_{i1}, p_{i2}, p_{i3}, \ldots, p_{in}]^T \tag{1}$$

In Equation 2, $S_i^t$ represents the velocity vector per repetition for particle $i$.

$$S_i^t = [s_{i1}, s_{i2}, s_{i3}, \ldots, s_{in}]^T \tag{2}$$

In Equation 3, $Obj1$ denotes the internal multiplication of $w$ on the vector of velocity.

$$Obj1_{ij} = wS_{ij}^t \tag{3}$$

Equation 3 affects the situation (here, we refer to velocity) of the vector in the next step. This means the distance between the two points (the first point refers to the solution of the problem, and the second point talks about the position of the particle) in the search space is highly dependent on the value of $w$ such that if we increase $w$; the search speed will increase, while the accuracy will decrease. However, this may lead us to obtain a more accurate solution in the next position.

$$Obj2_{ij} = c_1 Random_1^t(pBest_{ij} - P_{ij}^t) \tag{4}$$

Equation 4 is based on personal experience and self-perception of the particle. If the individual experience is slightly different from the current situation, it will point to a new location with a constant coefficient $c1$ indicating the effective value. In addition, a random variable $Random_1$ prevents the parameters from converging.

$$Obj3_{ij} = c_2 Random_2^t(gBest_j - P_{ij}^t) \tag{5}$$

Equation 5 refers to the best social experience. The result is the sharing of individual experiences. If the current position of the particle differs from the best social experience, it leads to a new position that has an impact factor of $c_2$. A random variable $Random_2$ prevents the convergence of the parameters.

In Equation 6, all three objects influence the velocity of the next step.

$$S_{ij}^{t+1} = Obj1_{ij} + Obj2_{ij} + Obj3_{ij} \tag{6}$$

In Equation 7, $P_{ij}^{t+1}$ points to the next position, where calculated by the sum of the current position and the obtained velocity.

$$P_{ij}^{t+1} = P_{ij}^t + S_{ij}^{t+1} \tag{7}$$

Indeed, the PSO algorithm has shown a suitable optimization approach for all types of problems with continuous and discrete data.

## 2.3 WHALE OPTIMIZATION ALGORITHM (WOA)

The algorithm was proposed by Mirjalili et al. in 2016 Mirjalili & Lewis (2016). It is developed upon the hunting mechanism of humpback whales in nature. As described in Mirjalili & Lewis (2016), whales have common cells in some regions of their brain, similar to humans. Therefore, they are capable of learning, judging, communicating, and becoming emotional. The hunting method of whales, the bubble-net feeding method, has been studied and it was found to be interesting Watkins & Schevill (1979). Distinctive bubbles along a circle similar to a '9'-shaped path are created. Further investigation Goldbogen et al. (2013) of whales' hunting method shows that two maneuvers are associated with the bubbles namely 'upward-spirals' and 'double-loops'. While the latter consists of three stages such as coral loop, lobtail, and capture loop; the former is created by the descent of whales to around 12 meters down and the creation of bubbles in a spiral shape around the prey and ascending of the whales towards the surface. The mathematical model of the WOA utilizes three models of encircling prey, spiral bubble-net feeding maneuver, and search for prey. A set of random solutions is assumed in the WOA algorithm. At each iteration, the positions of search agents are updated with respect to either a randomly chosen search agent or the best-obtained solution. In order to globally optimize the algorithm, exploration or exploitation abilities are included by decreasing the parameter a from 2 to 0.

## 3 DATA REPRESENTATION

In this section, we present the two data sets; Sensors Data (SD) and Repairs Data (RD), which are taken to carry out the proposed TDO approach.

The SD data includes the aggregated vehicle usage, where the values of the predictors/parameters are collected each time a vehicle visits an authorized workshop for repairs and service. These sensor data were collected from heavy-duty trucks designed and used in forests over two years of operation in China, from 2018 to 2020.

The RD includes information regarding the faults that were reported during the vehicle's operational life. In particular, the RD gives information about the vehicle, the part, and the failure date. We integrated these two data sets (using the approach introduced in Wu & Meeker (2002)) to build a complete data set having both usages (as independent parameters) and fault information (as dependent and target values). Each row of the data shows the behavior/usage of the vehicle in a specific duration of time (e.g., a week), and the target value indicates whether the vehicle is sound (0) or defective (1) given the usage.

The data includes 9541 samples collected over time and 300 parameters characterizing vehicle usage. To detect the failures (in this study, we focus on the part of the power train component), we processed and analyzed incredibly imbalanced data in which the proportion of the majority class ('1' healthy vehicles) is extremely higher than the minority class ('0' unhealthy vehicles).

## 4 PROPOSED APPROACH

In this section, we describe how our proposed tuple and feature selection approach can find the best representation of data for fault detection. We formulate this problem as an optimization task, where the classifier trains the model–iteratively–using a different subset of data to find out the portion that provides the best prediction performance.

The conceptual view of the proposed approach is illustrated in Figure 1, where at the first step, TDO randomly initializes the first population, including a set of individual solutions (chromosomes). A representation of a chromosome is shown in Figure 1, where each chromosome is divided into two parts; in the first part, the tuples are placed, and the second part holds the parameters. Given the first population (it may include several individual solutions), TDO calculates the fitness value to evaluate the performance of the generated solutions. Equation 8 describes how the fitness will be calculated, which includes four different objectives.

$$
\begin{aligned}
Fitness = W_1 \times Obj_1 + W_2 \times Obj_2 \\
+ W_3 \times Obj_3 + W_4 \times Obj_4
\end{aligned}
\tag{8}
$$

Where $W_i$ refers to the weight of the objectives in the fitness function–the sum of those weights should be equal to one shown in equation 9.

$$
W_1 + W_2 + W_3 + W_4 = 1
\tag{9}
$$

$Obj_1$ in Equation 10 expresses the performance of the classifier used to predict the fault. In this problem, the goal is to minimize the error, where $f(x)$ is converted into $1 - f(x)$.

$$
Obj_1 = 1 - f(x)
\tag{10}
$$

$$
Obj_2 = \frac{\sum_{i=1}^{n} Selected\_Tuple_i}{\sum_{i=1}^{n} Tuple_i}
\tag{11}
$$

$$
Obj_3 = \frac{\sum_{i=1}^{n} Selected\_Feature_i}{\sum_{i=1}^{n} Feature_i}
\tag{12}
$$

Equation 11 defines the second objective, which reduces the number of selected tuples. While the third objective defined in Equation 12 shows the number of selected features that should be diminished.

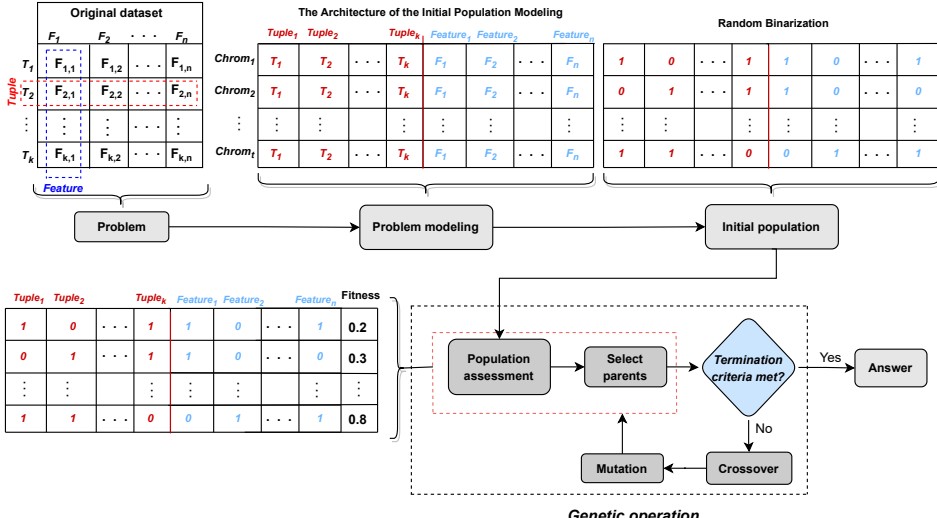

Figure 1: The conceptual view of the proposed approach- with. In this schema, we could observe how the input data are divided into train and test, then converted into chromosomes where tuples and features are positioned side by side.

$$Obj_4 = 1 - \frac{\sum_{i=1}^{n} Selected\_Tuple(Minority)_i}{\sum_{i=1}^{n} Tuple(Minority)_i} \qquad (13)$$

$Obj_4$ in Equation 13, indicates the number of minority tuples that should be increased. Since the objective function is decreasing, it is necessary to convert the whole equation to the negative power of one. Note: it needs to be mentioned that the above objectives are set in different ranges that should be optimized at the same time over the course of generations:

The TDO continuously calls the GA operators (such as selection, mutation, and crossover) to select the best solution and prepare it for the next generations. The optimization process will be terminated until the criterion is met, which is the maximum number of generations. The proposed approach selects the most informative tuples and features in each generation in order to increase the performance of the predictive models. Moreover, in this fashion, we could decrease the time consumption at reaching the best performance over the optimization process.

## 5 EXPERIMENTAL EVALUATION AND RESULTS

### 5.1 STUDY SETUP

As outlined in Section 1, the primary objective of this study is to develop a *fault detection* approach with a specific focus on addressing the challenges posed by highly imbalanced data in the automotive industry. Therefore, in order to conduct the experiments, we have defined two Experimental Goals (EGs) as follows:

- $EG1$: To what extent can we predict component failures based on the vehicle's usage data?

- $EG2$: How can we utilize the GA, PSO, and WOA algorithms for tuple and feature selection, and what represents the optimal trade-off between tuples and features?

The two aforementioned Experimental Goals (EGs) delineate our evaluation criteria, which are aligned with the primary objective of this study. Our intention was to address these questions by leveraging various structures and data sources, primarily focusing on classification. As a result, we conducted the following experiments:

| Parameters | V(s) |
|---|---|
| $Weight_1(W1)$ | 0.97 |
| $Weight_2(W2)$ | 0.01 |
| $Weight_3(W3)$ | 0.01 |
| $Weight_4(W4)$ | 0.01 |
| Number of generation | 150 |
| Size of population | 28 |
| Length of chromosome | 367 |
| Ratio of elite | 0.5 |
| Probability of crossover | 0.8 |
| Probability of mutation | 0.1 |
| Parents portion | 0.5 |
| The number of executions of the objective function | 2114 |

(a) GA

| Parameters | V(s) |
|---|---|
| $Weight_1(W1)$ | 0.97 |
| $Weight_2(W2)$ | 0.01 |
| $Weight_3(W3)$ | 0.01 |
| $Weight_4(W4)$ | 0.01 |
| $c_1, c_2$ | 1, 3 |
| $w$ | 0.9 |
| $k$ | 14 |
| $p$ | 5 |
| Num of generation | 151 |
| Size of particles | 14 |
| Length of Particles | 367 |
| The number of executions of the objective function | 2114 |

(b) PSO

| Parameters | V(s) |
|---|---|
| $Weight_1(W1)$ | 0.97 |
| $Weight_2(W2)$ | 0.01 |
| $Weight_3(W3)$ | 0.01 |
| $Weight_4(W4)$ | 0.01 |
| $b$ | 1 |
| Num of generation | 151 |
| Number of whales | 14 |
| number of Feature | 367 |
| The number of executions of the objective function | 2114 |

(c) WOA

Table 1: Objective values and parameters of three optimization algorithms. We take advantage of the hyper-parameter estimator available in scikit-learn: https://scikit-learn.org/stable/modules/grid$_s$earch.html to obtain the values.

## 5.2 EVALUATION AND RESULTS

Before answering the first EG, we conducted several experiments by building predictive models using different machine-learning algorithms. These experiments were developed to find a baseline (or set of baseline) to assess our proposed approach. In all experiments, the dataset, which contains 9511 instances, was divided into training sets with 7189 samples and test sets holding 352 samples testing the models. The figures illustrated in Table 2 show the performance of eleven algorithms on the dataset. It is quite evident that Xgboost and AdaBoost outperformed other algorithms with 0.94 and 0.84, respectively. Thus, we consider these numbers as our baseline to implement and assess our optimization approach.

Given the performance of XGboost Chen & Guestrin (2016), we utilized this algorithm at the core of our objective function. The Area Under the Curve (AUC) Kumar & Indrayan (2011) was used as the performance metric. To achieve the best results, we parameterized all three optimization algorithms using the values shown in Table 1 for GA, PSO, and WOA, respectively.

To ensure a fair comparison among the aforementioned algorithms, it's important that the number of executions for each individual objective function remains consistent. Hence, in the GA process, the objective function is called based on the number of chromosomes in the population. Initially, the function generates 28 populations, but in the subsequent generations, it will execute 14 populations. However, the entire process will run for 150 generations, as depicted in Figure 2. Based on the provided information, for PSO and WOA, the population sizes are multiplied by the number of iterations to ensure comparability with the GA settings.

During the under-sampling process, it was observed that in certain instances, the value of $AUC$ decreased while attempting to preserve the minority data. As a result, to achieve the highest AUC value, we parameterized the proportion of minority samples using $W_4$ in Equation 13 when constructing the models.

Figure 2 illustrates the AUC values obtained over multiple generations using GA, PSO, and WOA algorithms. We conducted three types of experiments, which included feature selection only, tuple selection only, and simultaneous tuple and feature selection, in order to evaluate the performance of the approach.

Regarding feature selection alone, WOA showcased superior performance compared to GA and PSO, achieving more than a 5% improvement overall, as illustrated in Figure 2(c). This outcome suggests the potential promise of WOA in tackling complex problems of this nature. However, when considering both tuple and feature selections (Figure 2(a and b)), as well as tuple selection alone, both GA and WOA ultimately yield similar levels of performance by the conclusion of the

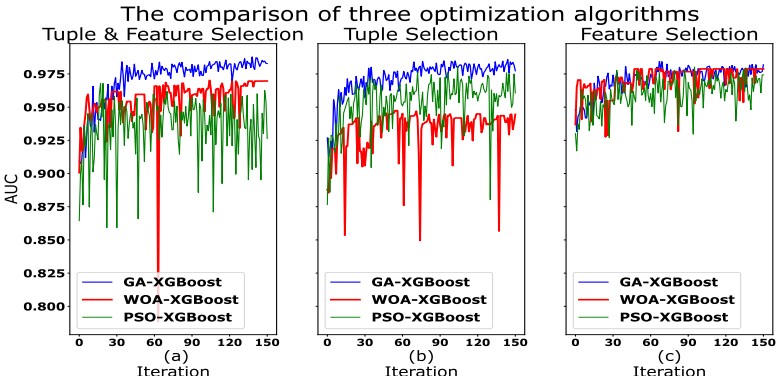

Figure 2: The comparison of three optimization algorithms.

| Method | Objects | | | | AUC | # of tuple | # of ftr | # of minr | t |
|---|---|---|---|---|---|---|---|---|---|
| | AUC | tuple | feature | minority | | | | | |
| SVM | | | | | 0.8323 | 7189 | 367 | 161 | 14.46 |
| ExtraTrees | | | | | 0.7230 | 7189 | 367 | 161 | 1.15 |
| GaussianProcess | | | | | 0.6960 | 7189 | 367 | 161 | 536.27 |
| KNeighbors | | | | | 0.7320 | 7189 | 367 | 161 | 1.64 |
| LGBM | | | | | 0.8815 | 7189 | 367 | 161 | 6.34 |
| Logistic Regression | | | | | 0.7714 | 7189 | 367 | 161 | 0.54 |
| QDA | | | | | 0.4998 | 7189 | 367 | 161 | 1.14 |
| RandomForest | | | | | 0.8035 | 7189 | 367 | 161 | 14.57 |
| SGD | | | | | 0.7314 | 7189 | 367 | 161 | 12.45 |
| XGBoost | | | | | 0.9446 | 7189 | 367 | 161 | 33 |
| AdaBoost | | | | | 0.8402 | 7189 | 367 | 161 | 12.28 |
| SMOTE+XGBoost | | | | | 0.9131 | 14056 | 367 | 7028 | 14.04 |
| **GA+XGBoost** | ✓ | ✓ | ✓ | ✓ | **0.9878** | 3176 | 152 | 67 | 7674 |
| GA+XGBoost | ✓ | ✓ | | ✓ | 0.9858 | 3043 | 367 | 72 | 21425 |
| GA+XGBoost | ✓ | | ✓ | | 0.9844 | 7189 | 174 | 161 | 30249 |
| PSO+XGBoost | ✓ | ✓ | ✓ | ✓ | 0.9679 | 3804 | 184 | 88 | 15042 |
| PSO+XGBoost | ✓ | ✓ | | ✓ | 0.9805 | 3642 | 367 | 79 | 28369 |
| PSO+XGBoost | ✓ | | ✓ | | 0.9802 | 7189 | 177 | 161 | 34329 |
| WOA+XGBoost | ✓ | ✓ | ✓ | ✓ | 0.9696 | 6487 | 329 | 143 | 28278 |
| WOA+XGBoost | ✓ | ✓ | | ✓ | 0.9445 | 2192 | 367 | 46 | 43219 |
| WOA+XGBoost | ✓ | | ✓ | | 0.9788 | 7189 | 183 | 161 | 19697 |

Table 2: Comparative table of the combination of methods and objectives. ftr expresses the number of features; minr refers to the minorities; and t points to the execution time per second.

optimization process. It's worth noting that the AUC values obtained using the PSO approach exhibit considerable variance over the course of 150 generations in all three experiments. Conversely, both GA and WOA show consistent improvement with the progression of generations, nearly reaching an AUC of 0.99% in tuple selection, as illustrated in Figure 2(b), and tuple and feature selections, as depicted in Figure 2(a).

Table 2 provides comprehensive information regarding the experiments, including the computational time required for the combined techniques. Notably, the results indicate that GA and XGboost outperformed all combinations, even in scenarios with a limited number of minority instances, specifically 67 and 72. Regarding the execution time, as indicated in Table 2, it is evident that the GA approach is significantly faster than WOA. This observation strongly suggests that when time efficiency is a critical factor, WOA may not be the ideal choice. Upon closer examination of the algorithms and the number of objectives to be optimized over the generations, it was observed that GA outperformed others, especially when considering four objectives. Conversely, when two objectives are considered (as shown in Figure 2c), GA and PSO exhibit similar performance.

Considering the figures reported in Table 2, the GA+XGBoost performed well compared with other combinations, including linear and optimizations. However, the superiority of our proposed approach to this problem was undeniable. This motivated us to assess all these approaches in different contexts. Thus, we conducted the proposed approach with the other approaches on two different datasets (African Country Recession Kaggle and SECOM UCI) to evaluate the generality and verify

| Algorithm | African Country Recession | | | SECOM | | |
|---|---|---|---|---|---|---|
| | F1-Score | t-test ** | p | F1-Score | t-test ** | p |
| XGBoost | 0.9133 | 30.722 ✓ | <0.05 | 0.9154 | 16.22077✓ | <0.05 |
| SVM | 0.8793 | 46.21362✓ | <0.05 | 0.9008 | 27.03977✓ | <0.05 |
| ExtraTrees | 0.9021 | 30.80877✓ | <0.05 | 0.9002 | 26.43019✓ | <0.05 |
| GaussianProcess | 0.8793 | 46.21362✓ | <0.05 | 0.9008 | 27.03977✓ | <0.05 |
| KNeighbors | 0.8793 | 46.21362✓ | <0.05 | 0.9008 | 27.03977✓ | <0.05 |
| LGBM | 0.9056 | 34.20818✓ | <0.05 | 0.9008 | 27.03977✓ | <0.05 |
| Logistic Regression | 0.9021 | 35.81065✓ | <0.05 | 0.8960 | 30.59082✓ | <0.05 |
| QDA | 0.8793 | 46.21362✓ | <0.05 | 0.9008 | 27.03977✓ | <0.05 |
| RandomForest | 0.8793 | 46.21362✓ | <0.05 | 0.9008 | 27.03977✓ | <0.05 |
| SGD | 0.8793 | 46.21362✓ | <0.05 | 0.9037 | 9.37029✓ | <0.05 |
| AdaBoost | 0.8885 | 42.00354✓ | <0.05 | 0.9070 | 22.48418✓ | <0.05 |
| SMOTE+XGBoost | 0.9131 | 40.77441✓ | <0.05 | 0.9234 | 10.30378✓ | <0.05 |
| **GA+Tuple+Feature+XGBoost** | **0.9808** | – | – | **0.9396** | – | – |

Table 3: The comparison between GA+XGBoost and other algotihms. For each model, $5 \times 2$ cv paired t-test was used to test the pairwise significance between the model and other models for each task. "**" refers to the alpha level at 0.05 to reject the null hypothesis, e.g., the "two sigma" level. Significant differences are denoted by ✓, and insignificant differences are denoted by ✗.

whether the GA+XGBoost performed equally or better than its performance in other problems. This is quite an important consideration since it evaluates the generality of the approach to deal with data from different domains.

Table 3 shows the implementation evaluation of different approaches on the two datasets, in which we can observe that GA+XGBoost outperformed other algorithms in both cases by an average f-score value of 0.98 and 0.93 for dataset 1 and dataset 2, respectively. In addition, we performed the statistical t-test and compared the results received for GA+XGBoost and all other experiments to quantify whether the outcomes differed significantly. Selecting $\alpha = 0.05$ as the critical value, we could see in all experiments that the test rejected the null hypothesis and concluded that the proposed approach performed best.

## 6 SUMMERY

In this preliminary work, we proposed a fault detection system designed for the automotive industry. We have developed a two-dimensional approach to data reduction employing optimization algorithms, enabling us to identify and extract the most informative components from the data for the construction of predictive models. Our approach aimed to map vehicle usage to component failures using optimization algorithms, with a specific focus on addressing and handling highly imbalanced data. We examined the GA, PSO, and WOA algorithms with the aim of simultaneously reducing both the tuple and feature space to facilitate the construction of predictive models. Furthermore, we employed a similar reduction approach by considering only features and types. The experimental results demonstrate the promise of the proposed technique for reducing data dimensions and suggest a high potential for further investigation. The generality experiments also show how the proposed optimization approach could perform in other contexts. However, more datasets (from different contexts) are needed to extensively assess this aspect of the approach. In future work, our goal is to explore a broader range of genetic algorithms and integrate them into a deep neural network framework to map vehicle usage to component breakdowns effectively.

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
