# OpenReview forum: "A Fault Forecasting Approach Using Two-Dimensional Optimization (TDO)"
_ICLR.cc/2024/Conference — Submitted to ICLR 2024_

### Official Review · Reviewer_SqTT · 2023-10-30

**Soundness:** 2 fair
**Presentation:** 2 fair
**Contribution:** 1 poor
**Rating:** 3
**Confidence:** 4

**Summary:**

In this paper, the authors address data preparation for fault detection within the automotive industry. Confronted with the challenge of high-dimensional feature spaces and imbalanced data, they introduce a two-dimensional optimization technique. Utilizing the Genetic Algorithm, they aim to concurrently reduce both data point tuples and the feature space. Additionally, the paper explores Particle Swarm Optimization (PSO) and Whale Optimization Algorithms (WOA).

Evaluating the GA, PSO, and WOA algorithms, they simplify tuple and feature space. The experimental results showed the method's capabilities in enhancing the classification performance.

**Strengths:**

Originality:
The paper introduces a two-dimensional optimization method for the application of fault detection in the automotive sector, by integrating the Genetic Algorithm with PSO and WOA. This approach offers the handling of high-dimensional data and imbalances.

Quality:
The paper is well-structured and methodologically sound clear, it presents the optimization techniques applied in the context of automotive fault detection.

Clarity:
paper is generally well written.

Significance:
The study achieves promising results. Observations regarding GA's speed compared to WOA have practical implications.

**Weaknesses:**

* **Lack of Comparative Analysis:**
While the paper does explore GA, PSO, and WOA for the automotive fault detection problem, it lacks a comparison with other existing methods or state-of-the-art techniques. Providing such a comparative study would have offered more context to the results. Also, adding references to the previous works/results in the domain would add to the paper results validation.

* **Dataset Concerns:**
The paper could have benefited from a more comprehensive analysis using multiple datasets or a more diversified dataset. Relying on a single dataset might limit the generalizability of the method and findings. Moreover, the paper lacks in describing the stats of the dataset in findings, it didn't provide comprehensive results that include the percentage/number of useful/selected features/tuples.


* **Parameterization Details:**
The authors mention parameterizing the three optimization algorithms as per values shown in "Table 1", yet they didn't explain how these optimal values were derived. Understanding the selection process for these parameters is essential for reproducibility and validation of the results.

To enhance the paper's standing, the authors could provide a broader comparative analysis by diversifying the datasets used, investigating the parameter selection, investigating PSO's behavior in more depth, and possibly expanding on the optimization objectives.

**Questions:**

1- Could you elaborate on the reason that the paper lacks a comparison with existing methods or state-of-the-art techniques related to fault detection in the automotive industry?

2- Could you elaborate on the reason that only a singular dataset was chosen for this study, especially given its number of features?

3- In the results section, there seems to be limited detailed statistics about the dataset. Could you provide more statistics, especially the percentage/number of useful features/tuples?

4- How were the optimal parameter values of the three optimization algorithms determined?

---

> ### Author Response · Authors · 2023-11-19
> **Response to the comments**
>
> Many thanks for the valuable comments. The four bullet points you mentioned are quite right and pretty crucial for our study. Thus, to answer your questions, we are running our approach on other datasets together with other ML algorithms to assess our approach in different contexts. We are updating the paper with a section to show the comparison results. We are also describing how we reached the optimal parameters illustrated in the tables.  Regarding the statistics about the dataset, we are pretty limited in providing such detailed information since the data is quite sensitive for the company we are collaborating with. However, we try to add more information that can answer your concerns and follow the company regulations.

---

### Official Review · Reviewer_5nRP · 2023-11-02

**Soundness:** 2 fair
**Presentation:** 2 fair
**Contribution:** 2 fair
**Rating:** 1
**Confidence:** 3

**Summary:**

This paper present a fault prediction method in automotives that uses optimizations on the recorded data of a vehicle and associate features. The selectionof the most appropriate data and features results in better predictions. For the data selection these optimization algorithms are used: Genetic Algorithm (GA) Whitley (1994), Particle Swarm Optimization (PSO) Marini & Walczak (2015), and Whale Optimization Algorithm (WOA) Mirjalili & Lewis (2016). The main ideas to to findout the relevant data that provides the better performance. Therefore, the above optimization algorithms are compared for that purpose.

**Strengths:**

a. The details of the use of GA for selection of data for fault prediction
b. Experiments comparing GA, PSO and WOA algorithms.

**Weaknesses:**

a. The details about optimization algorithms are not necessary, rather good references should be enough.
b. No details are provide on how the PSA and WOA are used in the proposed methods
c. No comparisons ar shown with state of the art (SOA)methods.
d. Nor the Data used is described neither the data source.

**Questions:**

a. Why only the details of GA applications are give but not the PSA and WOA?
b. Why the propsoed methods is not comapred with SOA methods?
c. What is the source of the data used?
d. What is an example of the data?

---

> ### Author Response · Authors · 2023-11-19
> **Response to the comment**
>
> Thank you for the comment. We are updating the paper with more explanation to answer the question.

---

> ### Comment · Reviewer_5nRP · 2023-11-21
> **MISTAKE CORRECTION**
>
> I accidentally entered the wrong (another paper) review. Now I have corrected it. I sincerely apologize for this mistake.

---

### Official Review · Reviewer_wBRJ · 2023-11-02

**Soundness:** 2 fair
**Presentation:** 2 fair
**Contribution:** 2 fair
**Rating:** 3
**Confidence:** 5

**Summary:**

The paper presents the relevant issue of training data imbalance and poses it as an optimization problem. A comprehensive set of heuristic techniques have been employed and results on XGboost are presented. The paper examines the GA, PSO,and WOA algorithms with the aim of simultaneously reducing both the tuple andf eature space to facilitate the construction of predictive models.

**Strengths:**

The comparison with several metaheusristics on a particular data set.

**Weaknesses:**

The importance of the fat-shattering dimension in explaining the beneficial effect of a large margin on generalization performance is discussed in existing literature (John Shawe-Taylor, Algorithmica, 22,157-172,1998; J John Shawe-Taylor, Peter Bartlett, Robert Williamson and Martin Anthony, IEEE Trans. Inf. Theory, 44 (5) 1926-1940, 1998. ). In the past, adaboost and thetaboost have been proposed to tackle this problem with theoretical results on maximum margin likelihood estimation under class imbalance. The problem statement is therefore not novel. Weighted loss functions are also proposed to solve this kind of optimization problems. The experimentation is not diverse. A carefully chosen data set is used. Since there is no theoretical guarantee of the method, it is difficult to ascertain the efficacy unless the experimentation net is widened to include some anomaly detection data sets.

**Questions:**

My question is how different is this approach except for superior performance on some some chosen data sets. Will this approach work if there is heavy imbalance (<1% in one class, rest in other classes) or SVM ensemble training?

---

> ### Author Response · Authors · 2023-11-19
> **Response to the comments**
>
> Thank you for the comments.  I think the question was quite important to show the generality of the approach. Thus, to answer your concerns, we are implementing our approach (and other machine learning algorithms) on different datasets to see how the approach performs for different problems. We will update the paper with these results.

---

### Official Review · Reviewer_gDy3 · 2023-11-02

**Soundness:** 2 fair
**Presentation:** 2 fair
**Contribution:** 2 fair
**Rating:** 3
**Confidence:** 4

**Summary:**

Interesting paper that focuses on developing specialised techniques for tackling the challenge of high-dimensionality in real-world industrial data on the face of class-imbalance. The authors use popular algorithms like GA, PSO etc. and integrate it with common ML models to evaluate it on a sensors and repairs dataset from China. The paper has limited novelty and the proposed methodology does not consider the explainability and trustworthiness aspects to a sufficient level for a safety-critical industrial application.

**Strengths:**

The topic is interesting, the use of real-world dataset for experiments is promising. This reviewer appreciates the utilisation of simpler ML models like XGBoost for experimental purposes.

**Weaknesses:**

Lack of baseline models is one of the major weaknesses of the paper. The authors only consider XGBoost for classification purposes and do not compare it with any other ML model. The dataset under consideration (about 10k samples) is quite small and the authors do not clearly mention the rationale behind specifically using XGBoost and not any other model for classification task. Another major issue this reviewer spotted is the lack of focus on explainability in the paper - GA+XGBoost or PSO+XGBoost is mentioned in the paper and a graph shown with the AUC, however, it is clearly not enough to be convinced of how the GA+XGBoost framework is actually working and learning in the background on the face of high dimensional data with high class imbalance.

**Questions:**

It would have been nice to see the authors to have performed additional experiments with more traditional and popular algorithms like SMOTE (which the paper does mention of in the introduction section), however, lack of comparison with any kind of baseline dimensionality reduction or class-balancing/synthetic data augmentation algorithm or ML model (other than XGBoost) makes the paper lack novelty and has limited contribution.

---

> ### Author Response · Authors · 2023-11-17
>
> Thank you for the valuable comments.
> We carefully review your comments and try to answer them.
> Thus, we have started to train different models to have more baselines to compare with our proposed approach.   We also agree that there is a lack of explanation to justify the machine learning approach we used, so we try to clarify and rationalize using our approach and the results we obtained. We are also implementing our proposed approach on other datasets to assess the generality of the approach. This will definitely strengthen the study by showing how general the proposed ML approach is. We will update the manuscript with the above points.
> We hope the new version will answer your concerns about the points.

---

> ### Author Response · Authors · 2023-11-21
> **Response to the comments**
>
> Thank you for the valuable comments you provided for our studies. We carefully reviewed the comments and attempt to answer most of them. From the comments, we found two major points that we needed to answer as follows:
> 1) The lack comparison with other ML algorithm:
> To answer this question, we run other ML algorithms and reported their results in Table 2.
>
> 2) The lack of implementing our approach on other datasets:
> To answer this question we also implement our approach and other ML algorithms on two more datasets to assess generality of the proposed approach.
>
> Regarding the data. As we mentioned, the data is from a company which a confidential and must anonymize the sensitive statistics so that why we couldn't report those numbers.
>
> We have edited and added more explanation how we reach the best parameters in our study. All the changes and information added in the paper have been highlighted in blue so you can see them.
>
> We hope the new implementations and clarifications could answer your concerns to accept the paper for the publication.
>
> Regards
> Authors

---

### Comment · Area_Chair_cGTe · 2023-11-21
**[Time Sensitive, ICLR24] Please read the authors' responses and try to discuss the remaining concerns with the authors**

Dear Reviewers,

The authors have provided detailed responses to your comments.

Could you have a look and try to discuss the remaining concerns with the authors? The reviewer-author discussion will end in two days.

We do hope the reviewer-author discussion can be effective in clarifying unnecessary misunderstandings between reviewers and the authors.

Best regards,

Your AC

---

### Meta-Review · Area_Chair_cGTe · 2023-12-06

**Metareview:**

The paper, while presenting a potentially interesting approach to automotive fault detection using ML algorithms, falls short in several critical areas. The lack of comparative analysis with state-of-the-art techniques, insufficient dataset diversity, and missing parameterization details significantly undermine the paper's contribution to the field. Furthermore, the exclusive focus on XGBoost without comparing it with other models, combined with a lack of explainability and theoretical novelty, limits the paper's academic value.

The authors also agreed that reviews make sense and would like to add more experiments to make it more solid. However, the follow-up experiments are not presented. Thus, given these significant shortcomings, it is recommended that the paper be rejected. Future revisions should address these concerns, particularly focusing on comparative analysis, dataset diversity, methodological transparency, and theoretical innovation to make a more substantial contribution to the field.

**Justification For Why Not Higher Score:**

The paper, while presenting a potentially interesting approach to automotive fault detection using ML algorithms, falls short in several critical areas. The lack of comparative analysis with state-of-the-art techniques, insufficient dataset diversity, and missing parameterization details significantly undermine the paper's contribution to the field. Furthermore, the exclusive focus on XGBoost without comparing it with other models, combined with a lack of explainability and theoretical novelty, limits the paper's academic value.

**Justification For Why Not Lower Score:**

N/A

---

### Decision · Program_Chairs · 2024-01-16

Reject